# A Cu-SiO$_2$ Catalyst for Highly Efficient Hydrogenation of Methyl Formate to Methanol

Jincheng Wu [1,2], Guoguo Liu [1,2,*] , Qin Liu [1,2], Yajing Zhang [1,2], Fu Ding [1,2] and Kangjun Wang [1,2,*]

1 College of Chemical Engineering, Shenyang University of Chemical Technology, Shenyang 110142, China; wujincheng0210@163.com (J.W.); liuqinlimeng@163.com (Q.L.); yjzhang2009@163.com (Y.Z.); dingfu@syuct.edu.cn (F.D.)
2 Research Center for Green Catalytic Materials and Process Technology and Engineering, Shenyang 110142, China
* Correspondence: lguoguo@syuct.edu.cn (G.L.); wangkj_dut@syuct.edu.cn (K.W.); Tel.: +86-024-8938-3760 (G.L.)

**Abstract:** The hydrogenation of methyl formate to methanol is considered one of the most effective methods for recycling methyl formate products. We recently developed a highly efficient and cost-effective Cu-SiO$_2$ catalyst using the ammonia-evaporation (AE) method. The Cu-SiO$_2$-AE catalyst demonstrated superior performance, achieving a methyl formate conversion of 94.2% and a methanol selectivity of 99.9% in the liquid product. The catalyst also displayed excellent stability over a durability test of 250 h. Compared to the commonly used Cu-Cr catalyst in the industry, the Cu-SiO$_2$-AE catalyst exhibited higher conversion of methyl formate and methanol yield under the same reaction conditions. Characterization results revealed a significant presence of Si-OH groups in the Cu-SiO$_2$-AE catalyst. These groups enhanced the hydrogen spillover effect and improved hydrogenation efficiency by preventing sintering during the reaction to stabilize the Cu species. The strategy employed in this study is applicable to the rational design of highly efficient catalysts for industrial applications.

**Keywords:** Cu-SiO$_2$ catalyst; hydrogenation of methyl formate; methanol; catalyst stability

## 1. Introduction

Methyl formate (MF) is a major by-product of converting syngas to ethylene glycol. However, it is highly unfavored in industrial engineering due to its low-boiling point, high flammability, and high transportation cost [1–3]. Consequently, it is highly desirable to explore methods for converting MF into value-added commodities. One such method is the hydrogenation of methyl formate to produce methanol, which is a vital platform molecule in the modern chemical industry. Moreover, methanol is widely recognized as a hydrogen carrier for fuel cells, further highlighting its significance [4,5].

The Cu-Cr-based catalyst widely utilized in the industry has shown low MF conversion efficiency, typically below 80% [6]. Furthermore, the inclusion of Cr in this catalyst poses environmental concerns and potential health risks for operators involved in its production and usage. Additionally, the presence of Cr significantly increases the cost associated with catalyst recycling [7–10]. On the other hand, conventional copper-based catalysts suffer from rapid deactivation due to metal sintering. The abundance of Cu$^0$ species in these Cu-based catalysts leads to the undesired decomposition of methanol into CO$_x$ under an H$_2$-rich atmosphere, resulting in reduced methanol selectivity [11,12]. Consequently, the productivity of methanol from MF using these catalysts is adversely affected.

To address the abovementioned issues, we have successfully developed a Cu-SiO$_2$-AE catalyst using the ammonia-evaporation (AE) method, which offers high efficiency and low cost. This catalyst demonstrated exceptional performance in the hydrogenation of MF to methanol. Operating under the mild reaction condition of 140 °C and 1.5 MPa, the

catalyst maintained a stable MF conversion of 94% with a methanol selectivity of over 99% during a durability test spanning 250 h. The Cu-SiO$_2$-AE catalyst was characterized by a significant presence of Si-OH groups, playing a crucial role in dispersing and stabilizing the Cu species during the reaction. This feature improved the catalyst resistance against sintering, reducing the formation of Cu$^0$ species responsible for methanol decomposition into CO$_x$. Consequently, the catalyst exhibited high activity, excellent selectivity, and durability [13,14]. Furthermore, the Si-OH groups promoted the generation of hydrogen spillover, greatly enhancing the concentration of hydrogen on the surface of the Cu-SiO$_2$-AE catalyst, thereby improving the MF conversion and overall performance.

## 2. Results and Discussion

### 2.1. Characterization of the Catalysts

The X-ray diffraction (XRD) patterns are present in Figure 1A for as-calcined and in Figure 1B for the spent Cu-SiO$_2$-CP, which was synthesized via the co-precipitation method and Cu-SiO$_2$-AE catalysts. The diffraction peak at ~22.5° observed corresponds to amorphous SiO$_2$ in the calcined catalysts [15]. The Cu-SiO$_2$-CP exhibits distinct peaks at 35.45°, 38.74°, and 48.74°, which are attributed to the diffraction of the CuO phase (JCPDS 45-0937) [16–18]. The diffraction peaks at 36.4°, 61.4°, and 69.6° shown in Figure 1B indicate the presence of the Cu$_2$O phase in the spent catalysts (JCPDS 34-1354). Moreover, the spent Cu-SiO$_2$-CP catalyst displays additional peaks at 43.3°, 50.5°, and 74.1°, which correspond to the metallic Cu phase (JCPDS 04-0836) [19]. Notably, the absences of diffraction peaks corresponding to the Cu$^0$ phase are absent in the Cu-SiO$_2$-AE catalysts, indicating that the Cu nanoparticles in the catalyst are highly dispersed with sizes smaller than 2 nm [20,21].

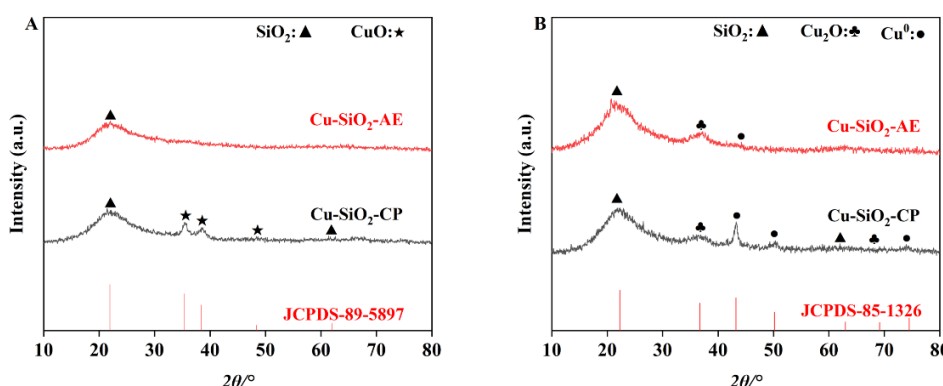

**Figure 1.** XRD patterns of (**A**) as-calcined and (**B**) spent Cu-SiO$_2$-CP and Cu-SiO$_2$-AE catalysts.

The pore structure of Cu-SiO$_2$-CP and Cu-SiO$_2$-AE catalysts were characterized by N$_2$ physisorption and the results are presented in Figure 2 and Table 1. As seen from Table 1, the Cu-SiO$_2$-AE catalyst exhibited a specific surface area up to 350.8 m$^2$/g. The Cu dispersion and surface area of Cu, crucial factors affecting the catalytic activity and stability of Cu-based catalysts, were measured by the N$_2$O titration. As shown in Table 1, the Cu-SiO$_2$-AE catalyst exhibited higher values of Cu dispersion (D) and Cu surface area (S$_{Cu}$) than the Cu-SiO$_2$-CP catalyst.

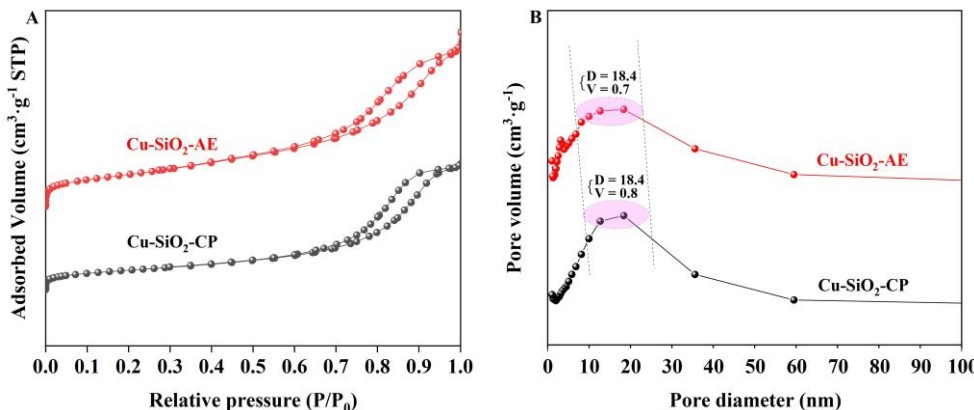

**Figure 2.** (**A**) Nitrogen sorption isotherms and (**B**) pore diameter distribution of the as-synthesized catalysts.

**Table 1.** Physicochemical properties of the catalysts prepared by different methods.

| Samples | [a] $S_{BET}$ (m$^2$/g) | [a] V (cm$^3$/g) | [a] d (nm) | [b] D (%) | [b] $S_{Cu}$ (m$^2$/g) | [b] $d_{Cu}$ (nm) |
|---|---|---|---|---|---|---|
| Cu-SiO$_2$-CP | 208.3 | 0.8 | 18.4 | 50.0 | 339.0 | 2.0 |
| Cu-SiO$_2$-AE | 350.8 | 0.7 | 18.4 | 109.6 | 742.9 | 0.9 |

[a] Determined by the N$_2$-adsorption method and [b] by the N$_2$O titration method.

Figure 3 shows the results of the temperature-programmed reduction test (H$_2$-TPR) using hydrogen gas. Both the Cu-SiO$_2$-CP and Cu-SiO$_2$-AE catalysts exhibit reduction peaks during the H$_2$-TPR analysis. The Cu-SiO$_2$-CP catalyst displays a peal at 235 °C, indicating the reduction of the CuO$_x$ phase to Cu. In contrast, the Cu-SiO$_2$-AE catalyst shows a peak at a lower reduction temperature of 212 °C. A lower reduction temperature indicates that the crystallite sizes are smaller on the metal surfaces of catalysts, which facilitates the diffusion of H$_2$ [22]. Therefore, it is believed that the Cu-SiO$_2$-AE has a high dispersion of Cu nanoparticles within the catalyst, leading to enhanced interaction between metal and support materials in the catalysis [23,24].

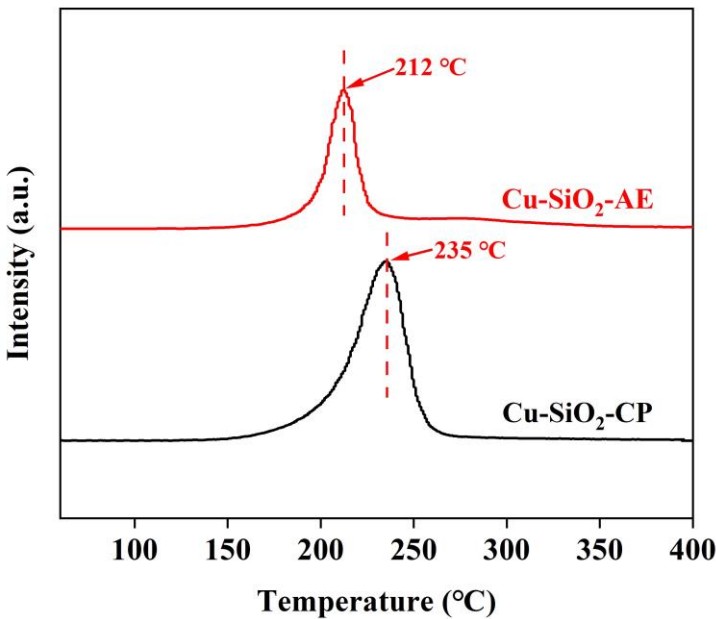

**Figure 3.** H$_2$-TPR profiles of the Cu-SiO$_2$-CP and Cu-SiO$_2$-AE catalysts.

Figure 4 depicts the transmission electron microscope (TEM) images of the reduced Cu-SiO$_2$-CP and Cu-SiO$_2$-AE catalysts. The results show that the average size of the Cu nanoparticles is 11.0 nm for the reduced Cu-SiO$_2$-CP catalyst, whereas it is significantly reduced to about 2.2 nm for the reduced Cu-SiO$_2$-AE catalyst [25]. The decrease in the Cu nanoparticle size indicates that the Cu-SiO$_2$-AE catalyst exhibits superior resistance to sintering during the reaction compared to the Cu-SiO$_2$-CP catalyst. The enhanced interaction between the Cu metal and SiO$_2$ support contributes to this improved sintering resistance, consistent with the results revealed by XRD and H$_2$-TPR analyses [26,27].

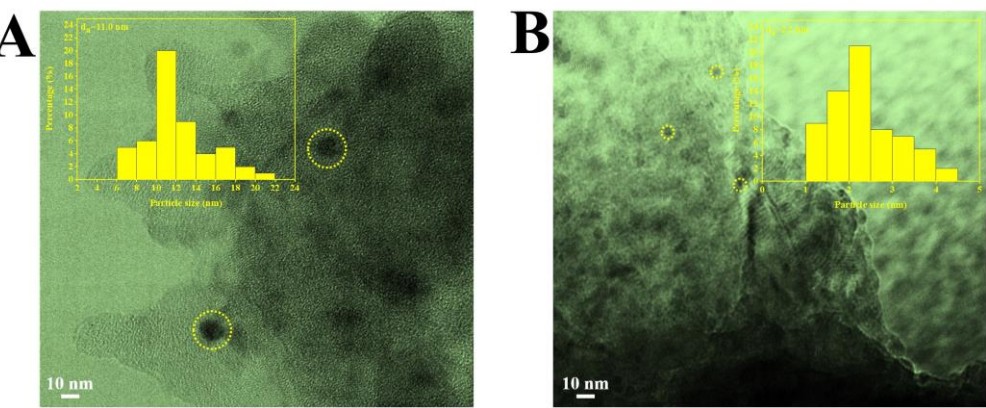

**Figure 4.** TEM images of reduced (**A**) Cu-SiO$_2$-CP and (**B**) Cu-SiO$_2$-AE catalysts with the pictures of Cu particle size distributions inset.

Figure 5 presents the X-ray photoelectron spectroscopy (XPS) spectra of the reduced and spent Cu-SiO$_2$-CP and Cu-SiO$_2$-AE catalysts. Prior to analyzing the Cu 2p spectra, we carried out the charge correction of C 1s spectra at 284.8 eV. In the Cu 2p spectra, no satellite peaks are detected, indicating the complete reduction of Cu species. Both the reduced catalysts display broad peaks centered for the Cu 2p core-level spectrum at 933.0 eV, corresponding to the coexistence of Cu$^0$ and Cu$^{\delta+}$ species. Additionally, a small, broad peak at 935.5 eV in the Cu 2p core-level spectrum suggests the presence of Cu$^{2+}$ species originating from the CuSiO$_3$ phase [28]. Comparing the Cu 2p$_{3/2}$ peak in the Cu 2p core-level spectrum of the Cu-SiO$_2$-AE catalyst to that of the Cu-SiO$_2$-CP catalyst, the latter shows a shift towards lower binding energy, indicating a higher abundance of Cu species in a lower valence state, such as metallic copper [29]. These results are consistent with the findings from XRD and TEM analyses.

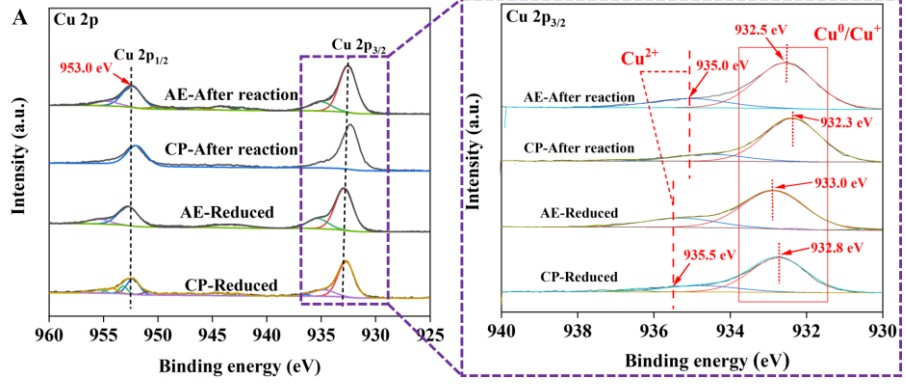

**Figure 5.** *Cont.*

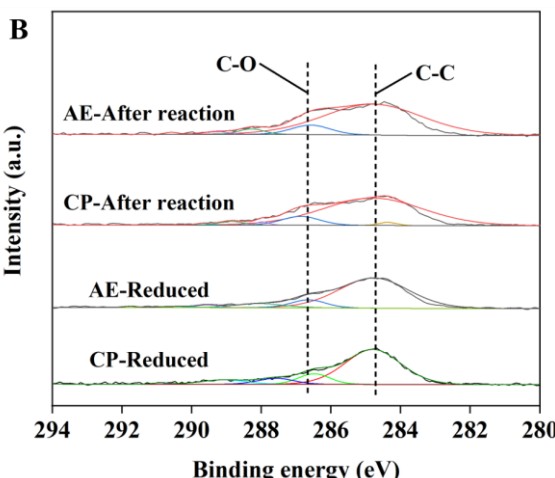

**Figure 5.** XPS spectra of the reduced and spent Cu-SiO$_2$-CP and Cu-SiO$_2$-AE catalysts of (**A**) Cu 2p and (**B**) C 1s.

The XPS spectra of the spent catalysts show similar patterns to the reduced catalysts, with a shift of 0.5 eV towards lower binding energy compared to the reduced catalysts, indicating a further reduction of Cu species during the hydrogenation of MF to methanol reaction [30,31].

### 2.2. Catalytic Hydrogenation of Methyl Formate

The catalytic performance of the Cu-SiO$_2$-CP and Cu-SiO$_2$-AE catalysts was evaluated in the hydrogenation of MF to methanol using a fixed-bed reactor. The reaction was conducted under controlled conditions: a temperature of 140 °C, a pressure of 1.5 MPa, and a liquid-hourly space velocity (LHSV$_{MF}$) of 2.4 h$^{-1}$. The feed gas comprised an H$_2$/MF molar ratio of 4:1. Table 2 and Figure 6 summarize the results of MF conversion, methanol selectivity, and methanol yield.

**Table 2.** Catalytic performance of the Cu-SiO$_2$-CP and Cu-SiO$_2$-AE catalysts in the hydrogenation of MF to methanol reaction.

| Catalyst | MF Conversion/% | Product Selectivity/% | | |
| --- | --- | --- | --- | --- |
| | | MeOH | DME [a] | Others [b] |
| Cu-SiO$_2$-CP | 83.9 | 92.1 | 5.7 | 2.2 |
| Cu-SiO$_2$-AE | 95.3 | 99.8 | 0.1 | 0.1 |

[a] By-product ethylene glycol dimethyl ether; [b] The sum of other over-hydrogenated by-products. Reaction conditions: LHSV$_{MF}$ = 2.4 h$^{-1}$, hydrogen-ester mole ratio = 4:1, reaction pressure = 1.5 MPa, reaction temperature = 140 °C, reaction time 9 h.

The results indicate that the Cu-SiO$_2$-CP catalyst achieves an MF conversion of 83.9% and a methanol selectivity of 92.1%. The gas phase by-products mainly consist of CO$_x$, while DME formation is negligible. There are some unknown products in the liquid. The CO$_x$ by-products are primarily generated through methanol decomposition. On the other hand, the Cu-SiO$_2$-AE catalyst exhibits superior performance with an MF conversion of 95.3% and a methanol selectivity of 99.8%, resulting in a methanol yield of 95.1%. The results highlight the superiority of the Cu-SiO$_2$-AE catalyst over the Cu-SiO$_2$-CP catalyst yielding higher conversion of MF and excellent selectivity towards methanol.

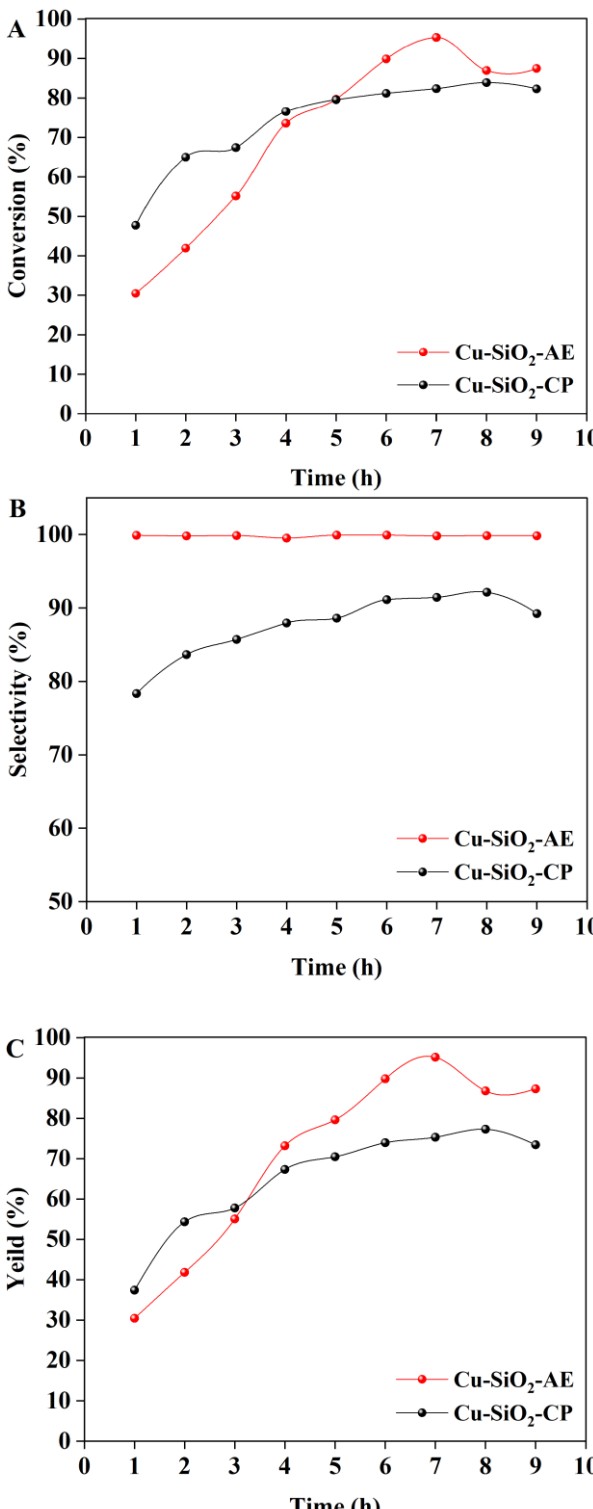

**Figure 6.** Catalytic performance of the Cu-SiO$_2$-CP and Cu-SiO$_2$-AE catalysts in hydrogenation of MF to methanol reaction of (**A**) MF conversion, (**B**) methanol selectivity, and (**C**) methanol yield. (Reaction conditions: LHSV$_{MF}$ = 2.4 h$^{-1}$, hydrogen-ester mole ratio is 4:1, reaction pressure at 1.5 MPa, reaction temperature of 140 °C).

In previous studies, we have investigated the catalytic performance of various catalysts in the hydrogenation reaction of MF to methanol. The results are summarized in Table 3. It is evident that the Cu-Cr and Cu-Cr-Mn catalysts perform unsatisfactorily [32], giving

MF conversions at 73.1% and 76.2%, respectively, under the reaction conditions of 140 °C, 0.1 MPa, and GHSV of 1800 mL gcat$^{-1}$ h$^{-1}$. The Cu-SiO$_2$ catalyst gave the MF conversion at only 51.5% under 180 °C at 0.5 MPa with LHSV of 10 h$^{-1}$. Though the addition of Mn could promote the catalytic activity, the MF conversion over Cu-Mn-SiO$_2$ was still unsatisfactory at 84.4% under equivalent conditions [33]. Moreover, the Cu-B$_2$O$_3$-SiO$_2$ catalytic even gave MF conversion at 51.5% under 150 °C at 2.8 MPa [34]. In contrast, the Cu-SiO$_2$-AE catalyst demonstrates superior performance, achieving an impressive MF conversion of as high as 95.3% and a methanol selectivity of nearly 100%.

**Table 3.** A summary of catalytic performance of MF hydrogenation to MeOH over copper catalysts.

| Catalyst | Cu Loading (wt%) | H$_2$/MF (*v/v*) | Temp. (°C) | P (MPa) | LHSV (h$^{-1}$) | MF Conv. (%) | MeOH Sel. (%) | Ref. |
|---|---|---|---|---|---|---|---|---|
| Cu-Cr | 23 | 4/1 | 140 | 0.1 | 1800 | 73.1 | 98.0 | [32] |
| Cu-Cr-Mn | 23 | 4/1 | 140 | 0.1 | 1800 | 76.15 | 99.55 | [32] |
| Cu-Mn-SiO$_2$ | 19.2 | 4/1 | 180 | 0.5 | 10 | 84.4 | 94.4 | [33] |
| Cu-SiO$_2$ | 19.2 | 4/1 | 180 | 0.5 | 10 | 51.5 | 90.4 | [33] |
| Cu-B$_2$O$_3$/SiO$_2$ | - | 3/1 | 150 | 2.8 | - | 39 | 99.8 | [34] |
| Cu-SiO$_2$-AE | 15 | 4/1 | 140 | 1.5 | 2.4 | 95.3 | 99.8 | This study |
| Cu-SiO$_2$-CP | 15 | 4/1 | 140 | 1.5 | 2.4 | 83.9 | 92.1 | This study |

We conducted the hydrogenation of MF to methanol reactions using the Cu-SiO$_2$-CP and Cu-SiO$_2$-AE catalysts over a temperature range of 120–160 °C. As illustrated in Figure 7B, the Cu-SiO$_2$-AE catalyst achieves an MF conversion of 70.3% and selectivities of 82.6% for methanol, 7.4% for DME, and 10% for other by-products at the reaction temperature of 120 °C. Increasing the temperature to 140 °C, the MF conversion and methanol selectivity increase significantly to 95.3% and 99.8%, respectively. However, when the reaction temperature is further increased to 160 °C, the catalytic activity of the Cu-SiO$_2$-AE declines, characterized by reductions in the MF conversion to 81.1% and the methanol selectivity to 86.2%. A similar tendency is also observed for the Cu-SiO$_2$-CP catalyst, although the latter shows a significantly low catalytic activity under the same reaction conditions. Therefore, the optimal temperature for Cu-SiO$_2$-CP and Cu-SiO$_2$-AE catalysts is 140 °C.

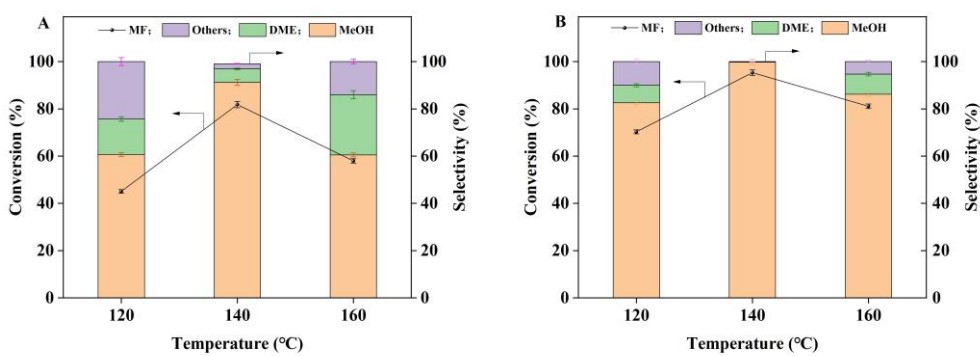

**Figure 7.** Catalytic performance of (**A**) Cu-SiO$_2$-CP catalyst and (**B**) Cu-SiO$_2$-AE catalyst at different reaction temperatures. (Reaction conditions: LHSV$_{MF}$ = 2.4 h$^{-1}$, hydrogen-ester mole ratio = 4:1, reaction pressure = 1.5 MPa).

Furthermore, we investigated the effect of reaction pressure on the hydrogenation of MF to methanol using the Cu-SiO$_2$-AE and Cu-SiO$_2$-CP catalysts with reaction pressure ranging from 0.1 to 2 MPa. At ambient pressure, the Cu-SiO$_2$-AE catalyst yields an MF conversion of 50.4%, with corresponding selectivities for methanol, DME, and others of 75.1, 15.8, and 9.1%, respectively (Figure 8B). Increasing the reaction pressure to 1.5 MPa significantly elevates the MF conversion and methanol selectivity to 95.3% and 99.8%, respectively. However, further raising the reaction pressure to 2.0 MPa leads to a decline in

the catalytic performance, reducing the MF conversion to 75% and the methanol selectivity to 88.3%. The Cu-SiO$_2$-CP catalyst exhibits a similar trend but with catalytic activity under the same reaction conditions. Consequently, the optimal reaction pressure for both the Cu-SiO$_2$-CP and Cu-SiO$_2$-AE catalysts is determined to be 1.5 MPa.

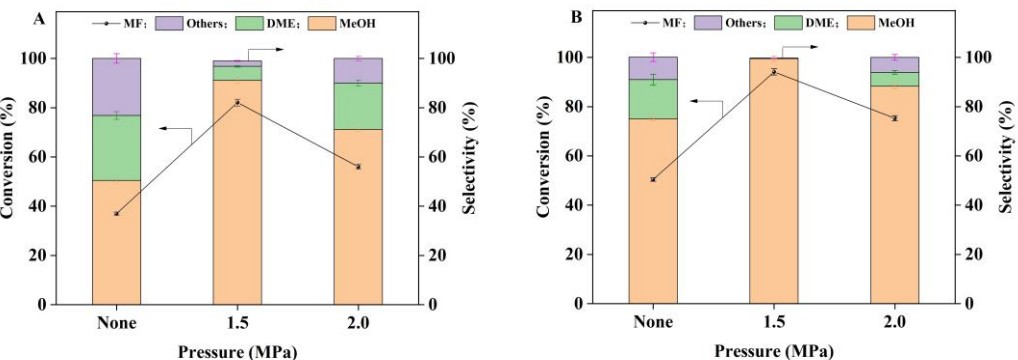

**Figure 8.** Catalytic performance of (**A**) the Cu-SiO$_2$-AE catalyst and (**B**) the Cu-SiO$_2$-CP catalyst at different reaction pressures. Reaction conditions: LHSV$_{MF}$ = 2.4 h$^{-1}$, the hydrogen-ester mole ratio of 4:1, and the reaction temperature of 140 °C.

Moreover, we conducted a durability test to evaluate the performance of the catalyst over a period of 250 h in the hydrogenation of MF to methanol at the optimized reaction conditions. Figure 9 depicts the results of MF conversion and methanol selectivity. As shown in Figure 9, the Cu-SiO$_2$-AE catalyst demonstrates a gradual increase in the MF conversion over the course of the durability test, starting from approximately 86% and reaching 95% after 250 h of continuous operation. Notably, the methanol selectivity remains constantly high at 99% in the liquid product throughout the entire test duration. These results indicate the excellent stability of the Cu-SiO$_2$-AE catalyst in the hydrogenation of MF to methanol reaction.

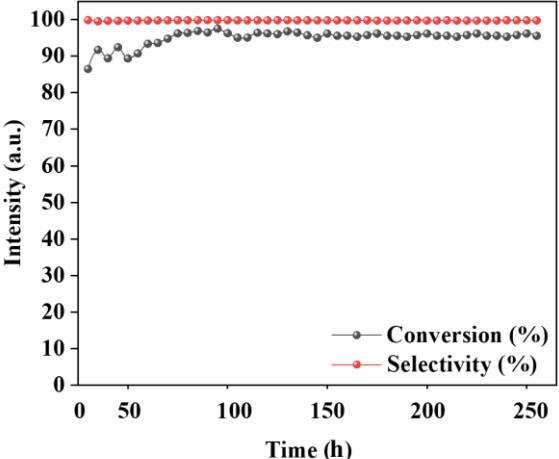

**Figure 9.** The durability test of Cu-SiO$_2$-AE catalyst in the hydrogenation of MF to methanol for 250 h. Reaction conditions: LHSV$_{MF}$ = 2.4 h$^{-1}$, the hydrogen-ester mole ratio of 4:1, the reaction pressure of 1.5 MPa, and the reaction temperature of 140 °C.

In addition, we conducted a comparison between the Cu-SiO$_2$-AE and the industrial Cu-Cr catalysts in the hydrogenation of MF to methanol reaction. As shown in Figure 10, both catalysts yield similar methanol selectivity, ranging from 98% to 99% in the liquid products. However, there is a notable difference in the MF conversion between the two catalysts. The Cu-SiO$_2$-AE catalyst demonstrates a remarkable increase in the MF conversion, from approximately 30% to 95%, within nine hours of reaction time. Conversely, the

industrial Cu-Cr catalyst delivers a constantly low MF conversion of only 60% under the same reaction conditions. These results confirm the superior catalytic performance of the Cu-SiO$_2$-AE catalyst in the hydrogenation of MF to methanol compared to the industrial Cu-Cr catalyst.

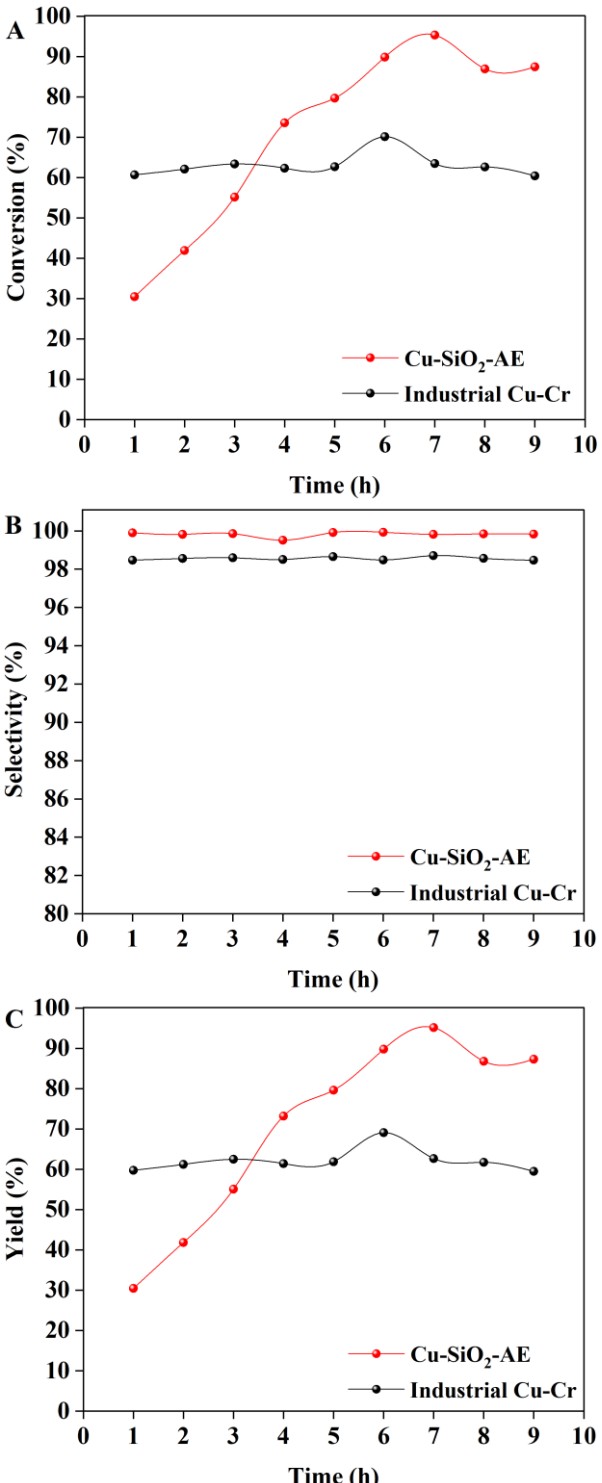

**Figure 10.** Comparison of the Cu-SiO$_2$-AE and industrial Cu-Cr catalysts in the hydrogenation of methyl formate to methanol reaction of (**A**) conversion of MF, (**B**) selectivity of methanol, and (**C**) yield of methanol. Reaction conditions: LHSV$_{MF}$ = 2.4 h$^{-1}$, the hydrogen-ester mole ratio of 4:1, the reaction pressure of 1.5 MPa, and the reaction temperature of 140 °C.

To elucidate the reaction mechanism of the hydrogenation of MF to methanol using the Cu-SiO$_2$-CP and Cu-SiO$_2$-AE catalysts, we carried out pyridine adsorption (Py-ADS) and pyridine desorption (Py-DES) Fourier transform infrared spectrometer (FT-IR) analyses. These analyses aimed to identify the presence of Lewis acid sites and Brønsted acid sites on the reduced and spent Cu-SiO$_2$-CP and Cu-SiO$_2$-AE catalysts [35]. As seen in Figure 11, the analyses reveal distinctive bands at approximately 1449 and 1540 cm$^{-1}$, which correspond to the Lewis acid sites and Brønsted acid sites, respectively. Additionally, the bands at 1489 and 1610 cm$^{-1}$ signify the coexistence of the Lewis acid sites and Brønsted acid sites [36]. Through careful analysis of the Py-ADS and Py-DES FT-IR spectra, we observe the presence of bands at approximately 1449, 1489, 1540, and 1610 cm$^{-1}$ in both catalysts, confirming the existence of both Lewis acid sites and Brønsted acid sites [37]. Notably, the intensities of these bands are more pronounced in both reduced and spent Cu-SiO$_2$-AE catalysts compared to the Cu-SiO$_2$-CP catalyst. This observation indicates that the Cu-SiO$_2$-AE catalyst possesses a higher density of acidic sites, including Brønsted acid sites (Si-OH) [38–40].

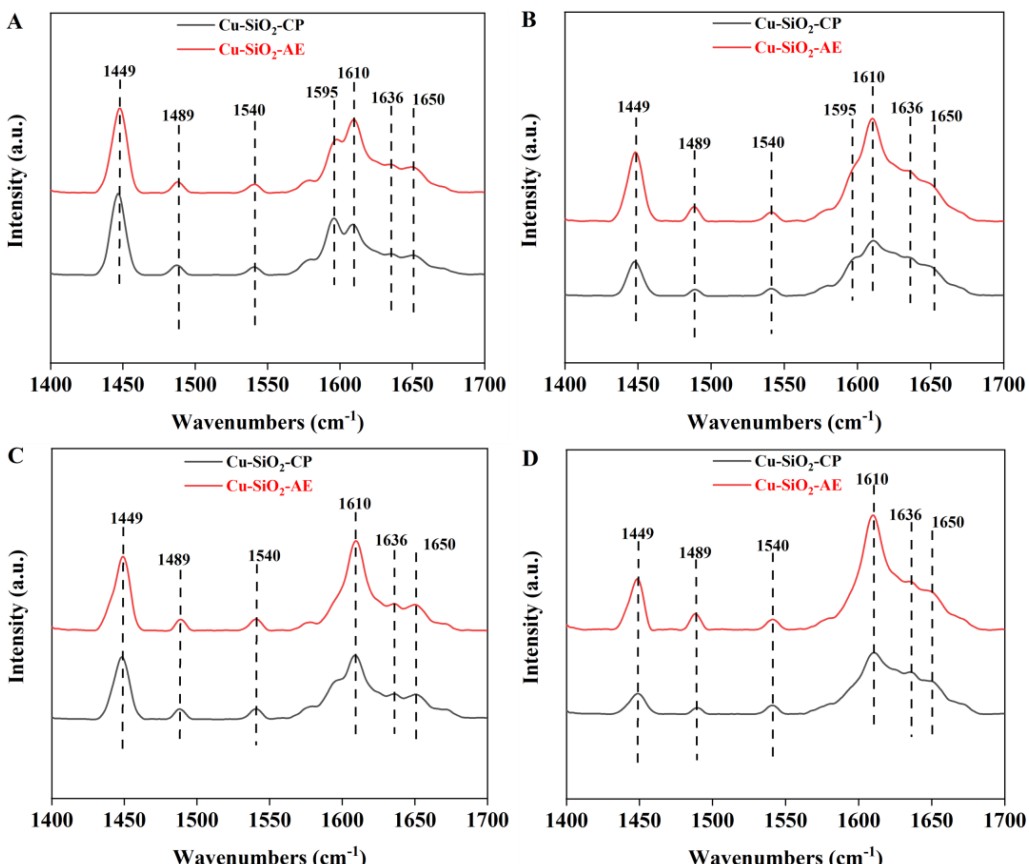

**Figure 11.** Pyridine–infrared spectra of Cu-SiO$_2$-CP and Cu-SiO$_2$-AE catalysts. (**A,C**) Reduced catalysts, (**B,D**) spent catalysts, (**A,B**) ADS, and (**C,D**) DES.

Figure 12 shows the FT–IR spectra obtained for the Si-OH over the Cu-SiO$_2$-CP and Cu-SiO$_2$-AE catalysts [41,42]. In the spectra, the bands observed at 3740, 3670, and 3646 cm$^{-1}$ are attributed to vibrations of the Si-OH groups [43,44]. Upon comparison with the Cu-SiO$_2$-CP catalyst, the Cu-SiO$_2$-AE catalyst shows higher intensity for these bands, indicating the presence of a greater number of Si-OH groups on the Cu-SiO$_2$-AE catalyst [45].

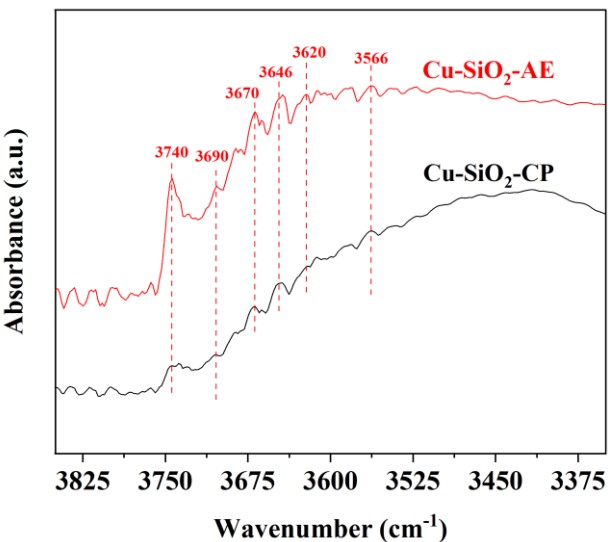

**Figure 12.** FT–IR spectra of Cu-SiO$_2$-CP and Cu-SiO$_2$-AE catalysts.

　　　There is a hypothesis that the Si-OH species contribute to the dispersion and stabilization of Cu nanoparticles during the calcination, reduction, and reaction processes. This phenomenon leads to increases in the Cu$^{\delta+}$ species and sintering resistance [46–49]. To verify this hypothesis, we used the TEM technique to measure the size of the Cu nanoparticles in the spent Cu-SiO$_2$-CP and Cu-SiO$_2$-AE catalysts. As expected, the size of the Cu nanoparticles in the spent Cu-SiO$_2$-CP catalyst is about 14 nm (Figure 13a), whereas it measures 2.4 nm in the Cu-SiO$_2$-AE catalyst (Figure 13b). The significant difference in the size of Cu nanoparticles suggests that the Cu-SiO$_2$-AE catalyst possesses higher sintering resistance than the Cu-SiO$_2$-CP catalyst [50,51].

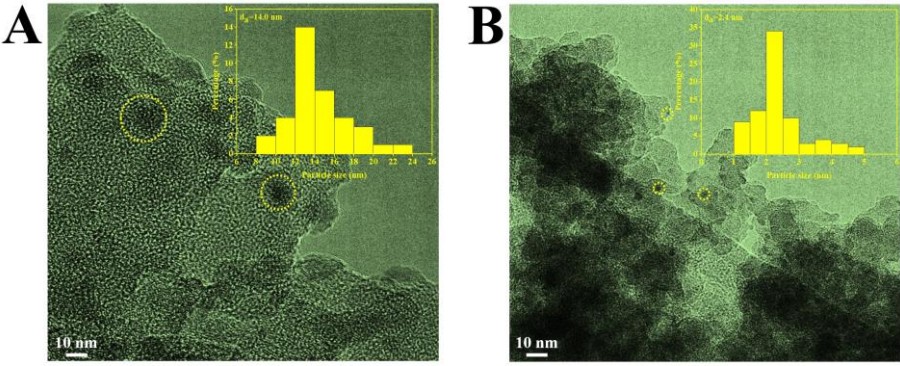

**Figure 13.** TEM images of (**a**) spent Cu-SiO$_2$-CP and (**b**) Cu-SiO$_2$-AE catalysts.

　　　To further investigate the catalytic behavior, we conducted a methanol decomposition reaction. The results depicted in Figure 14 demonstrate that the methanol decomposition decreases from 25% to 20% over 6 h with the Cu-SiO$_2$-CP catalyst. Conversely, the Cu-SiO$_2$-AE catalyst exhibits a consistently low methanol decomposition of approximately 14%. Since Cu$^0$ species act as the main active sites for methanol decomposition to CO$_x$, which is the main by-product of methanol decomposition in the gas phase, the observed decrease in methanol decomposition confirms a reduction in the Cu$^0$ species and an increase in Cu$^{\delta+}$ species [52,53]. In contrast, the constantly low methanol decomposition observed with the Cu-SiO$_2$-AE catalyst indicates a sustained presence of Cu$^0$ species, which is consistent with the XRD and TEM results, showing a higher proportion of Cu$^{\delta+}$ species and smaller Cu nanoparticle size in the Cu-SiO$_2$-AE catalyst. This behavior contributes to higher methanol

selectivity and enhanced sintering resistance, thereby improving methanol selectivity and sintering resistance [54–56].

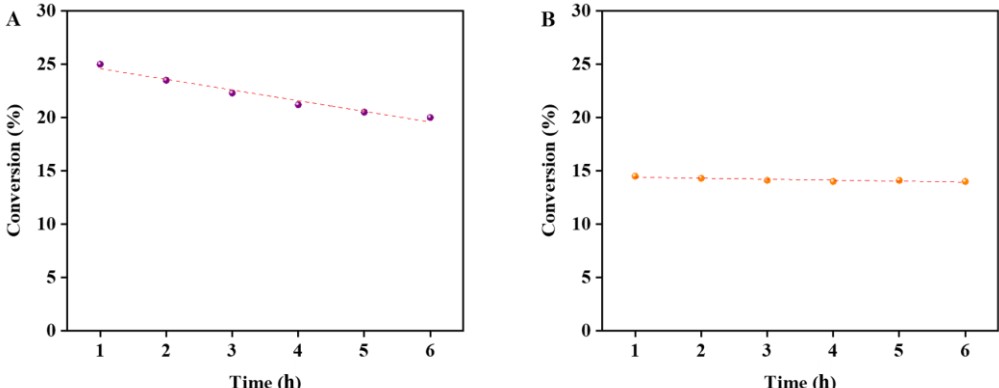

**Figure 14.** Catalytic performance of (**A**) Cu-SiO$_2$-CP and (**B**) Cu-SiO$_2$-AE catalysts in the decomposition of methanol reaction. Reaction conditions: 0.1 Mpa, 240 °C, 6000 mL g$^{-1}$ h$^{-1}$, MeOH/H$_2$/N$_2$ = 2/10/88 vol%.

In addition to the dispersion and stabilization of Cu nanoparticles, the Cu-SiO$_2$-AE catalyst benefits from the abundant Si-OH species, which promote hydrogen spillover, leading to a higher concentration of hydrogen on the catalyst surface. This increased hydrogen concentration has a significant impact on the catalytic activity in the hydrogenation of MF [57–59]. To investigate this effect, we conducted the hydroformylation of MF to methanol using feeds with various H$_2$ to MF molar ratios, including 1:1, 2:1, and 4:1. Figure 15 summarizes the results. For the Cu-SiO$_2$-CP catalyst, the MF conversion rates are 31.4%, 56.2%, and 83.9% for H$_2$/MF molar rations of 1:1, 2:1, and 4:1, respectively. Comparatively, the Cu-SiO$_2$-AE catalyst delivers higher MF conversion, particularly with the H$_2$/MF molar ratio of 4:1 under the same reaction conditions. Notably, the MF conversions are comparable between the Cu-SiO$_2$-CP catalyst with H$_2$/MF at 4:1 and the Cu-SiO$_2$-AE catalyst with the feed of H$_2$/MF at 2:1. This observation can be attributed to the influence of surface hydrogen concentration, which plays a crucial role in the hydrogenation of MF. The Cu-SiO$_2$-AE catalyst has a higher concentration of hydrogen on its surface facilitated by the abundant Si-OH species, which contributes significantly to its enhanced ability to hydrogenate MF to methanol [60,61].

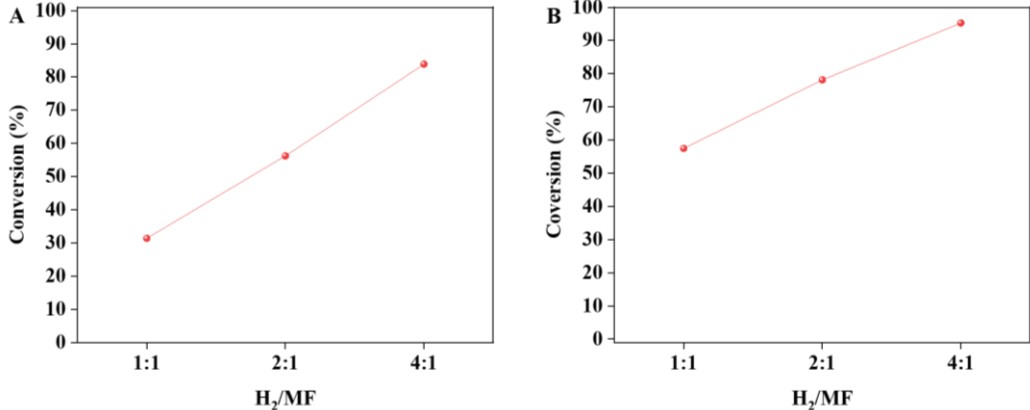

**Figure 15.** Catalytic performance of the (**A**) Cu-SiO$_2$-CP and (**B**) Cu-SiO$_2$-AE catalysts in hydrogenation of methyl formate to methanol reaction under the varied molar ratio of H$_2$/MF. Reaction conditions: LHSV$_{MF}$ = 2.4 h$^{-1}$, the hydrogen-ester ratios of 1:1, 2:1, and 4:1, reaction pressure of 1.5 MPa, and the reaction temperature of 140 °C.

## 3. Materials and Methods

### 3.1. Materials

Methyl formate (MF), and silica gel (30%) were purchased from Aladdin Chemical Reagent Company (Shanghai, China). Copper nitrate trihydrate [$Cu(NO_3)_2 \cdot 3H_2O$] (with a copper content of 15 wt%), ammonia solution ($NH_3 \cdot H_2O$, 28 wt%), ethanol, and urea were obtained from Sinopharm Chemical Reagent Co., Ltd. (Beijing, China). Feed gases containing pure hydrogen and nitrogen were supplied by Shenyang Hongsheng Gas Co., Ltd. (Shenyang, China).

### 3.2. Synthesis of Cu-SiO$_2$

Synthesis of Cu-SiO$_2$ by co-precipitation (Cu-SiO$_2$-CP). To synthesize Cu-SiO$_2$-CP, a typical run involved dissolving 30.37 g of $Cu(NO_3)_2 \cdot 3H_2O$ and 40 g of silica gel in distilled water under stirring. The resulting solution was slowly added to a beaker while vigorously stirring at room temperature, with the addition of a 1 M $Na_2CO_3$ solution. The pH value of the mixture was maintained at around 9.0. The precipitates were aged at room temperature for three hours, filtered, and washed five times with deionized water. The obtained precipitate was dried at 120 °C for 12 h and subsequently calcined in air at 500 °C for 4 h.

Synthesis of Cu-SiO$_2$ by ammonia-evaporation (Cu-SiO$_2$-AE). For the synthesis of Cu-SiO$_2$-AE, 30.37 g of $Cu(NO_3)_2 \cdot 3H_2O$ was dissolved in 100 mL of deionized water, and 28% ammonia aqueous solution was slowly added to the solution. The mixture was stirred for 30 min to allow for the formation of a copper ammonia complex solution. Subsequently, 40 g of silica gel was added to the copper ammonia solution and stirred for another 30 min. The suspension was then heated at 80 °C to allow for evaporation of ammonia until the pH value dropped to 6~7, while the copper species were deposited on silica. The resulting precipitate was filtered and washed five times with deionized water, dried at 120 °C for 12 h, and calcined in air at 500 °C for 4 h.

### 3.3. Catalyst Characterization

The textural parameters of the catalysts were determined using $N_2$ sorption isotherms measured with a BELSORP-miniX instrument (MicrotracBEL Japan, Inc. BELSORP-mini, Osaka, Japan). Before the measurements, all samples were pretreated at 200 °C under vacuum for ten hours to remove the surface contaminants. X-ray diffraction (XRD) patterns were obtained using a Rigaku Ultimate IV analyzer (Rigaku Japanese neo-Confucianism) with the radiation source of Cu Kα. The scanning rate was set at 10°/min in the range of 5~80°, with a voltage of 40 KV and a current of 15 mA. Transmission electron microscopy (TEM) images were acquired using an FEI Talos 200X microscope (Japan electronics JEM 2100F). Cu dispersion and the temperature-programmed reduction of hydrogen ($H_2$-TPR) were performed on the BELCAT-B apparatus (MicrotracBEL Japan, Inc.) equipped with a thermal conductivity detector (TCD) (MicrotracBEL Japan, Inc.). X-ray photoelectron spectroscopy (XPS) analysis was conducted using a PHI 1600 ESCA spectrometer (American Perkin-Elmer company, Waltham, MA, USA). Titration of acid pyridine-infrared spectroscopy (pyridine FT-IR) (Shanghai Lairui Scientific Instrument Co., Ltd., Shanghai, China) was used to determine the acid-site concentrations of the catalyst. This analysis utilized pyridine as the probe molecule and was carried out on a Thermo IS50 analyzer (Shanghai Lairui Scientific Instrument Co., Ltd.).

### 3.4. Catalytic Reaction

The catalytic hydrogenation of MF was performed in a continuous-flow fixed-bed reactor (Zhi Xiang Lan Tian Instruments Inc., Beijing, China). In a typical experiment, 2.0 mL of the as-prepared catalyst was placed in the middle of the reactor. The catalyst was initially reduced in a flow of 50 vol% $H_2/N_2$ (40 mL/min) at 250 °C for four hours. Subsequently, pure hydrogen (120 mL/min) and MF (0.08 mL/min) were introduced into the reactor at a pressure of 1.5 MPa and a temperature of 140 °C to initiate the reaction. The

liquid products were cooled, captured, and analyzed offline using a gas chromatograph with a flame ionization detector (FID, Zhe Jiang Fuli Analytical Instruments Inc. 7920Plus, Hangzhou, China).

The catalytic methanol decomposition reaction was conducted using a continuous-flow fixed-bed reactor. In a typical experiment, 0.5 g of the as-prepared catalyst was mixed with 2.0 g of quartz sand and positioned in the middle of the reactor. The catalyst was pre-reduced by passing a flow of 50 vol% $H_2/N_2$ mixture (40 mL min$^{-1}$) through at 250 °C for four hours. After the reduction step, the gas flow was switched to a mixture of methanol, $N_2$, and $H_2$ at 250 °C and ambient pressure to initiate the reaction. The gaseous products generated during the reaction were continuously analyzed using an online gas chromatograph equipped with a flame ionization detector (FID, Zhe Jiang Fuli Analytical Instruments Inc. 7920Plus, Hangzhou, China).

The MF conversion, products selectivity, and yield of methanol were calculated according to the following formulas:

$$X_{(MF)} = (n_{in(MF)} - n_{out(MF)})/n_{in(MF)} \times 100\% \tag{1}$$

$$S_{(MeOH)} = n_{out(MeOH)}/(n_{in(MF)} - n_{out(MF)}) \times 100\% \tag{2}$$

$$Y_{(MeOH)} = X_{(MF)} \times S_{(MeOH)} \tag{3}$$

$$X_{(MeOH)} = (n_{in(MeOH)} - n_{out(MeOH)})/n_{in(MeOH)} \times 100\% \tag{4}$$

where n is the mole rate (mol h$^{-1}$) of MF and MeOH.

## 4. Conclusions

In summary, the present study demonstrated the superior catalytic performance of the Cu-SiO$_2$ catalyst synthesized via the ammonia-evaporation method in the hydrogenation of MF to methanol. The catalyst exhibited a remarkable MF conversion of 95.3% and methanol selectivity of 99.8% in the liquid product. Furthermore, the catalyst displayed excellent durability over a 250-h test, surpassing the performance of the industrial Cu/Cr catalyst. Mechanistic investigations revealed that the Cu-SiO$_2$-AE catalyst possessed significant Si-OH groups, which played a crucial role in enhancing hydrogen spillover on the catalyst surface. This spillover effect contributed to the improved hydrogenation performance by increasing the concentration of hydrogen on the active sites.

Moreover, the presence of Si-OH groups facilitated the generation of Cu$^{\delta+}$ species, leading to higher methanol selectivity by preventing over-hydrogenation of methanol to CO$_x$. Additionally, the Si-OH played a vital role in stabilizing the Cu nanoparticle and inhibiting their sintering during the reaction. This resulted in a highly stable catalytic performance over the extended reaction times. The findings from this study provide valuable insights for catalyst design in the industry, particularly for hydrogenation reactions involving methyl formate. The strategy of using Si-OH groups to stabilize Cu nanoparticles holds great potential for developing efficient and durable catalysts in industrial applications.

**Author Contributions:** Conceptualization, J.W. and G.L.; methodology, J.W. and F.D.; formal analysis, J.W. and Y.Z.; data curation, J.W. and Q.L.; writing—original draft preparation, J.W.; supervision, G.L. and K.W. All authors have read and agreed to the published version of the manuscript.

**Funding:** This research was funded by the Natural Science Foundation of Liaoning Province (2021-NLTS-12-01, 2021-NLTS-12-03), Foundation of State Key Laboratory of High-efficiency Utilization of Coal and Green Chemical Engineering (2021-K64), Basic Research Project Educational Department of Liaoning Province (LJKZ0439, LJKQ2021068), Liaoning Innovation Talents Program in University (Liao [2020] 389), Shenyang Young and Middle-aged Science & Technology Talents Program (RC210365), Liaoning Revitalization Talents Program (XLYC1907029).

**Data Availability Statement:** Not applicable.

**Acknowledgments:** The authors would like to acknowledge the testing platform of the Analytical & Testing Center, Shenyang University of Chemical Technology for their help with characterizations.

**Conflicts of Interest:** The authors declare no conflict of interest.

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
