# Peer review of "A Cu-SiO2 Catalyst for Highly Efficient Hydrogenation of Methyl Formate to Methanol"

_catalysts, doi:10.3390/catal13071038_

Round 1

Reviewer 1 Report

The manuscript needs to be enriched with more scientific discussions with supporting results. The catalyst characterization results are not discussed sufficiently. Below are my detailed comments:

·         Mentioning Cu-SiO2-AE in the abstract, without clearly mentioning what AE is, might be confusing for the readers. As most readers read the abstract first.

·         Provide a citation for the effect of Cu0 species on the reduction of methanol to COx (line 35-38)

·         Provide evidence or citation on the effect of the Si-OH groups to disperse and stabilize the Cu nanoparticles. The authors mentioned that Si-OH prevents the sintering of the nanoparticles, but the XRD of the spent catalysts shows clear evidence of catalyst sintering.

·         Please mention at the beginning of the manuscript what is Cu-SiO2-AE and CP.

·         How was the H2-TPR done? Was the signal detected using a TCD or mass spec? Particles with smaller sizes tend to reduce at higher temperatures. Hence, why is the Cu-SiO2-AE reaction temperature lower than the Cu-SiO2-CP?

·         In line (85-86), authors wrote: “However, it was only ~2.1 nm 85 over the reduced Cu-SiO2-AE catalyst [20], suggesting that the Cu-SiO2-AE catalyst has a better sintering resistance during the reaction, due to the enhanced interaction between Cu metal and SiO2 support”. How is small particle size a claim for better sintering resistance, show the TEM image of the spent catalyst to prove this claim. The authors have mentioned in several instances the strong interaction between Cu and SiO2. Please provide concrete evidence for this claim. The XPS BE on Cu in both catalysts is similar, indicating that there was no significant interaction between the metal and support.

·         Provide TEM images with better resolution and higher magnifications.

·         Show the XPS charge correction done with the C1s spectra.

·         For the activity test the Cu particle size and dispersion is different for both the catalyst. Hence it is not fair to compare the activity without ToF. Please report the TOF of the catalyst to make the companion of catalyst activity. Otherwise, results in their present form are showing the effect of particle size in conversion.

·         Why was the activity CuCr, CuMn, and the catalyst from this work different? Please explain, with ToF and catalyst characterization results.

·         Provide error bars for the effect of temperature and pressure data.

·         How was the FTIR measurement done, the scale is missing. Why is the Si-OH functional group missing from the Cu-SiO­2-CP catalyst?

Author Response

Dear Reviewer,

      First of all, we thanks for your comments and suggestions. According to your nice suggestions, we have made extensive corrections to our previous manuscript.

      We feel sorry for our poor writings, however, we do invite a friend of us who is a native English speaker from the Canada to help polish our article. And we hope the revised manuscript could be acceptable for you.

Reviewer 2 Report

The manuscript is devoted at disclosing a novel effective Cu/SiO2 catalyst for the hydrogenation of methyl formate to methanol, exploiting the ammonia-evaporation synthesis method, using also a co-precipitated Cu/SiO2 catalyst as reference. The manuscript includes a set of characterization data that document the improved dispersion of the catalyst obtained by the AE route,  along with an increased concentration of hydroxyl groups likely favoring  hydrogen spillover phenomena. Overall such factors account for the superior activity-selectivity pattern of the AE catalyst in comparison to the reference and a commercial Cu-Cr catalysts, probed by considerably higher methanol yields larger than 90%, showing also a high stability during 250 h of r.o.t.. Overall the manuscript describes a study carried out with a systematic methodological approach, including a set of data that account for the main conclusions of the work. However, a rough English form and several syntax errors strongly hinder clarity and readibility of the manuscript. Therefore, the manuscript must be fully revised taking also into acocunt the following major inconsistencies and weaknesses:

- The data in tab. 1 concerning Cu dispersion, surface area and particle size seem not consistent. In particular, it is rather unbelievable that dispersion values of 0.3-1.3% would correpsond to particle size in the range of 0.8-2.9 nm.

- which the reason for the increasing conversion trend during the first 10h of reaciton with no changes in selectivity? this should be explained.

- the caption to Tab. 3 is not correct.

- Fig. 7 and Tab. 3 include the same information. In my opinion Fig. 7 could be omitted including in tab. 3 also the Cu loading.

- which the Cu loading of the studied catalysts also in comparison to literature data?

- A difference of 8°C in the maximum temperature of TPR peaks is not, in my opinion, so relevant to account for the different properties of AE and CP catalysts;

- the reaction time of the data shown in tab. 2 must be specified;

- the effects of T and P on the volcano-shaped trend of conversion and selectivity should be discussed;

- the spectra resolution in Fig. 13 is quite poor;

- at last, considering the similar catalyst formulations included in the literature analysis (Tab. 3), which the reasons for the different reactivity? 

The manuscript includes several sentences affected by syntax errors

Author Response

(The authors gave the same response as above.)

Reviewer 3 Report

The manuscript titled “Design a Cu-SiO2 catalyst for highly efficient hydrogenation of 2 methyl formate to methanol” presented by Jincheng Wu, Guoguo Liu, Qin Liu, Yajing Zhang, Fu Ding, and Kangjun Wang deals with the catalytic measurements and characterization of copper catalysts for methyl formate conversion to methanol. However, there are issues that could be clarified:

  • Line 40: Cu-SiO2-AE. What is AE in the name of catalysts? It is not explained.
  • Introduction should be re-written. It looks that some words are missed (for example, lines 25-26 conversion syngas to ethylene glycol).
  • Line 52: Cu-SiO2-CP is not explained, what CP means.
  • The abbreviations that are firstly mentioned should be explained, for example, XRD (line 52), CP, Ae, TEM, etc.
  • Table 1 contents non-understandable title of columns.
  • Line 79-81: The statement “probably due to the high dispersion of Cu nanoparticles, which would further enhance the interaction between metal and support over Cu-SiO2-AE catalyst” should be discussed more carefully and clearly.
  • The present of XPS study is badly written and should be revised. For example, “Both reduced catalysts gave broad peaks located at 932.98 eV”, the catalysts could not give any peaks, but the Cu2p core-level spectrum has a peak at 932.98 eV. The accuracy of binding energy is doubt. It should be 933.0 eV. Etc.

There is major issue concerning the manuscript: author believe that lower concentration of Cu0 than Cuδ+ and large amount of Si-OH lead to higher hydrogen spillover and MF to methanol conversion. But the hydrogen spillover needs the presence of metallic sites and non-metallic support. The sinter of Cu nanoparticle during the reaction looks doubt. It should be noted that both catalysts during the synthesis are calcined at 500C when the reaction process carried out at 140C.

The presentation of study is poor and should be revised. The current version should be rejected.

Missing words, the use of incorrect term. Incorrect use of verb tenses

Author Response

(The authors gave the same response as above.)

Round 2

Reviewer 1 Report

No further comments

Author Response

Dear Review,

Thanks for your comment. 

We have made further effort to improve the clarity and the readability of the manuscript. The some sentences and grammatical errors in the manuscript were thoroughly revised (noted by red color).

Reviewer 2 Report

The revised version of the manuscript takes into account most of my previous remarks and criticisms, representing an advancement with respect to the original submission. However, in my opinion, the authors should make a further effort to improve the clarity and the readability of the manuscript. Moreover, the data in Table 3 must be discussed in more detail considering that the performance of the various catalysts is compared at very different LHSV. 

The authors should make a further effort to render more fluent the English form throughout the manuscript

Author Response

(The authors gave the same response as above.)

Reviewer 3 Report

The authors took into account  my and the other reviewers comments that improves the manuscript.

Some sentences should be revised.

Author Response

(The authors gave the same response as above.)
